N6-methyladenosine (m6A) modification in inflammation: a bibliometric analysis and literature review

Li Zewen 1 2
Lao Yongfeng 1 2
Yan Rui 1 2
Guan Xin 1 2
Bai Yanan 1 3
Li Fuhan 1 2
Dong Zhilong 1 dzl19780829@163.com
1 Department of Urology, The Second Hospital of Lanzhou University, Lanzhou University , Lanzhou, Gansu , China
2 The Second Clinical Medical College, Lanzhou University , Lanzhou, Gansu , China
3 Laboratory Medicine Center, Lanzhou University , Lanzhou, Gansu , China
Uversky Vladimir
Electronic publication date: 2024 Dec 13
Publication date: 2024
Volume: 12
Electronic Location ID: e18645
Received 2024 Jul 18; Accepted 2024 Nov 14
Copyright: © 2024 Li et al.
Copyright year: 2024
Copyright holder: Li et al.
License: This is an open access article distributed under the terms of the Creative Commons Attribution License, which permits unrestricted use, distribution, reproduction and adaptation in any medium and for any purpose provided that it is properly attributed. For attribution, the original author(s), title, publication source (PeerJ) and either DOI or URL of the article must be cited.
License URL: https://creativecommons.org/licenses/by/4.0/

Keywords: Inflammation, m6A, Mechanism, Treatment, Bibliometrics

Funding: The authors received no funding for this work.

==============================
N6-methyladenosine (m6A) is the most abundant internal messenger RNA modification in eukaryotes, influencing various physiological and pathological processes by regulating RNA metabolism. Numerous studies have investigated the role of m6A in inflammatory responses and inflammatory diseases. In this study, VOSviewer and Citespace were used to perform bibliometric analysis to systematically evaluating the current landscape of research on the association between m6A and inflammation. The literature was sourced from the Web of Science Core Collection, with characteristics including year, country/region, institution, author, journal, citation, and keywords. According to the bibliometric analysis results of keywords, we present a narrative summary of the potential mechanisms by which m6A regulates inflammation. The results showed that the key mechanisms by which m6A modulates inflammation include apoptosis, autophagy, oxidative stress, immune cell dysfunction, and dysregulation of signaling pathways.

Introduction

Inflammation is a vital and evident protective response in organisms, serving as an essential adaptation to the disruption of cellular and tissue homeostasis. It plays numerous critical physiological roles, including host defense, tissue remodeling and repair, and metabolic regulation (Medzhitov, 2008). The inflammatory response is controlled by a complex regulatory network, with gene expression regulation being one of the key mechanisms of inflammatory control (Medzhitov & Horng, 2009).

RNA modification is a critical component of epigenetic regulation. More than 100 chemical modifications have been identified in cells, dynamically regulating the function and stability of RNA (Nachtergaele & He, 2018). N6-methyladenosine (m6A) is the most prevalent internal modification in eukaryotic mRNA, first identified in the early 1970s (Dubin & Taylor, 1975). Methylated RNA immunoprecipitation sequencing (MeRIP-seq) provided a comprehensive view of m6A modifications in human transcripts, revealing more than 10,000 m6A sites in over 25% of transcripts (Dominissini et al., 2012; Meyer et al., 2012). m6A is distributed across the transcriptome, being especially concentrated in long exons, near stop codons, and in the 3′ untranslated region (3′UTR). The 5′UTR and regions near the start codon also show varying levels of m6A modification (Zhou et al., 2015; Luo et al., 2014). m6A is a methylation modification that occurs at the N6 position of adenosine, which occurs predominantly in the RRACH sequence (where R = A or G and H = A, C or U) (Arribas-Hernández et al., 2021). m6A is a dynamic and reversible process (Batista, 2017). Its reversible modification is mainly dependent on three m6A-regulatory proteins, including the “writers” (methyltransferase), “erasers” (demethylase) and “readers” (m6A recognition proteins) (Shi, Wei & He, 2019). The primary m6A methyltransferase complex, composed of METTL3, METTL14, and Wilms’ tumor 1-Binding Protein (WTAP), catalyzes m6A formation, with METTL3 as the catalytic core, METTL14 as the RNA binding platform, and WTAP as the regulatory subunit that stabilizing the interaction between METTL3 and METTL14 and recruits METTL3/METTL14 heterodimers into nuclear speckles (Wang et al., 2016; Ping et al., 2014). Additional regulatory proteins associated with the complex include vir-like m6 A methyltransferase-associated protein (VIRMA, also known as KIAA 1429), RNA-binding motif protein 15/15 B (RBM 15/RBM 15 B), the E3 ubiquitin ligase HAKAI (also known as CBLL1) zinc-finger CCCH-containing structural domain protein 13 (ZC 3 H13), phosphorylated CTD interaction factor 1Phosphorylated CTD interaction factor 1 (PCIF1) (Yue et al., 2018; Patil et al., 2016; Bawankar et al., 2021; Wen et al., 2018; Wu et al., 2023). Besides mRNAs, m6A modifications also occur in ribosomal RNAs (rRNAs). METTL5 and ZCCHC4 are responsible for m6A modification of 18S rRNA and 28S rRNA, respectively (Ignatova et al., 2020; Ma et al., 2019). METTL16 is responsible for m6A modification of U6 spliceosomal small nuclear RNA (snRNA) (Pendleton et al., 2017). Demethylation of m6A is regulated by two “erasers,” FTO and ALKBH5, both members of the AlkB family of Fe(II)/α-KG-dependent dioxygenases. The human fat mass and obesity-associated protein (FTO) was the first RNA identified demethylase primarily catalyzing oxidative demethylation (Berulava et al., 2013; Fu et al., 2013), while ALKBH5 specifically demethylates single-stranded RNA (ssRNA) containing m6A (Zheng et al., 2013). The m6A readers bind to m6A-modified mRNAs and recruit various regulatory and functional mechanisms, ultimately influencing the fate of these target mRNAs. These readers include proteins containing YT521-B homology (YTH) structural domains (YTHDF1-3, YTHDC1-2), heterogeneous nuclear ribonucleoprotein C/G/A2B1 (HNRNPC/G/A2B1), insulin-like growth factor two mRNA-binding protein 1-3 (IGF2BP1-3), Leucine-rich PPR motif-containing protein (LRPPRC) and Proline-rich coiled-coil 2A (PRRC2A) and FMRP translation regulator 1 (FMR1) (Shi, Wei & He, 2019).

m6A is involved in almost all processes of mRNA metabolism, including RNA transcription, translation, and degradation. It alters the expression of target genes, thereby affecting the corresponding cellular processes and physiological functions (Roundtree et al., 2017; Liu et al., 2022a). Studies have shown a close relationship between m6A modifications and inflammatory responses to disease (Luo, Xu & Sun, 2021; Xu et al., 2022).

Bibliometric analysis examines literature characteristics systematically to qualitatively and quantitatively explore the research status and trends within a given field (Ma et al., 2021a). Specifically, bibliometrics assesses the status, trends, and frontiers of research activities by analyzing the basic characteristics of publications such as authors, countries/regions, institutions, disciplines, journals, and keywords and their intrinsic links (Wei et al., 2022). This methodology has become a prominent techniques for evaluating academic influence, quality, and credibility (Ellegaard & Wallin, 2015). Common bibliometric visualization tools such as VOSviewer and CiteSpace facilitate data analysis and visualization, enhancing reader comprehension (van Eck & Waltman, 2010; Pan et al., 2018).

To date, no bibliometric analysis has focused on m6A’s role in inflammatory responses. Addressing this gap, this study utilizes data from the Web of Science Core Collection (WoSCC). Relevant bibliometric data (including annual papers, countries/regions, authors, institutions, disciplines, journals, references, and keywords) in the field of m6A and inflammation research were searched for descriptive statistics. Based on the results, we discuss the current research status, key topics, and research frontiers on m6A’s involvement in inflammation, as well as potential mechanisms of m6A regulation in inflammatory responses.

Materials and Methods

Date collection

The Web of Science is one of the most widely utilized academic databases, serving as the primary research platform for hard sciences, social sciences, arts, and humanities information, as well as functioning as an independent global citation database for the most reputable publishers (Liu et al., 2022b). Additionally, the Web of Science is regarded as the optimal database for conducting bibliometric analyses (Ding & Yang, 2022). To enhance the representativeness and accessibility of the data, the Web of Science Core Collection (WOSCC) was chosen as the data source for this investigation. Furthermore, to ensure the precision of the retrieved data, the index selected was the Science Citation Index Expanded (SCI-expanded). The search terms employed were: (TS = (inflammatory OR inflammation OR inflammations)) AND (TS = (N-methyladenosine OR N(6)-methyladenosine OR N(6)mAdo OR N6-methyladenosine OR N6-methyladenosine (m6A) OR 6-methyladenosine)). The retrieved data were collected on December 31, 2023, to mitigate any potential discrepancies arising from daily updates. Following the aforementioned search strategy, 417 documents were initially retrieved from WoSCC. After excluding certain publication types, including meeting abstracts, editorial materials, early access, and retracted publications, 409 publications were retained, comprising 336 original research articles, and 73 review articles. The comprehensive screening process is illustrated in Fig. 1.

Figure 1 Flowchart of the search strategy and selection process.

Date analysis

The acquired text was processed using VOSvivewer software (version 1.6.19) and CiteSpace software (version 6.2.4) for subsequent analysis. VOSviewer facilitates the construction and visualization of bibliometric networks, including countries/regions, institutions, authors, journals, and keywords. CiteSpace was employed to investigate the clustering and bursting of keywords. In addition, the keyword timeline view was generated through CiteSpace, effectively illustrating the evolution of knowledge and the historical breadth of literature within a specific cluster over time, thereby enhancing our understanding of the developmental trajectory and trends within the field. For a more precise representation of the data, Scimago Graphica Beta 1.0.36 and GraphPad Prism 8.3.0 were also utilized for literature visualization.

Results

Analysis of annual publications distribution

The volume of publications serves as an indicator of the developmental trajectory and level of interest within the research domain. As depicted in Fig. 2, the literature pertaining to m6A and inflammation-related studies has been present since 1996. However, research in this domain remained sparse until 2018. Beginning in 2019, there has been a notable exponential increase in both the number of publications and citations, culminating in a total of 155 publications in 2022. This trend signifies a growing interest in this research area over the past 5 years.

Figure 2 Trends in the growth of publications and the number of citations.

Bibliometric analysis of countries/regions and institutions

These publications originated from 23 countries and 442 organizations/institutions. The leading five countries were China (n = 362, 88.51%), the United States (n = 40, 9.78%), Germany (n = 7, 1.71%), Italy (n = 5, 1.22%), and Canada (n = 4, 0.98%) (Table 1). The contributions from China and the United States constituted a significant majority of the total (98.29%). We subsequently visualized these 23 countries and developed collaborative networks based on the volume of publications and inter-country relationships (Fig. 3A).

Table 1 Top 5 productive countries/regions in m6A and inflammation.

Rank	Country	Record count	Citations	Average citation/Publication	
1	China	362	10,132	27.98895028	
2	USA	40	1,529	38.225	
3	Germany	7	219	31.28571429	
4	Italy	5	136	27.2	
5	India	4	64	16	

Figure 3 The visualization of countries regions (A) and institutions (B).

The size of nodes and edges is weighted by the number of published articles. The colors of the nodes represent different clusters. Generated by Scimago Graphica Beta 1.0.36.

The 10 institutions with the highest publication counts are detailed in Table 2. The top ten institutions were all situated in China, with the five institutions producing the most publications being Central South University (n = 26, 6.36%), Sun Yat-sen University (n = 26, 6.36%), Nanjing Medical University (n = 22, 5.38%), Shanghai Jiao Tong University (n = 22, 5.38%), Fudan University (n = 16, 3.91%). We visualized and constructed a collaborative network of 38 institutions that had a publication volume of five or more (Fig. 3B). While Central South University holds the record for the most publications, Shanghai Jiao Tong University is recognized as the most cited institution (n = 1,456) and demonstrates the closest collaborative ties with other institutions (total link strength = 18).

Table 2 Top 10 institutions in m6A and inflammation.

Rank	Organization	Record count	Times cited	Total link strength	
1	Central South University	26	451	4	
2	Sun Yat Sen University	26	1,236	15	
3	Nanjing Medical University	22	650	15	
4	Shanghai Jiao Tong University	22	1,456	18	
5	Fudan University	16	1,067	9	
6	Tongji University	15	405	9	
7	Zhengzhou University	13	352	4	
8	Zhejiang University	12	601	5	
9	Anhui Medical University	11	354	4	
10	China Medical University	10	198	0	

Bibliometric analysis of authors and co-cited authors

These publications involved 2,891 authors, with the top 10 authors ranked by article production listed in Table 3. Chuan He from The University of Chicago, Wei Wang from Sun Yat-sen University, and Qiong Xu from Sun Yat-Sen University Hospital of Stomatology emerged as the most prolific authors (n = 7). Qiong Xu is also the most cited author (n = 590). Among the 12,720 co-cited authors, seven authors were co-cited more than 100 times. The most frequently cited author was Xiang Wang (n = 266), followed by Kate D Meyer (n = 165) and Dan Dominissini (n = 125) (Table 4). Figures 4A and 4B illustrate the collaborative networks of authors and co-cited authors, respectively.

Table 3 Top 10 authors in m6A and inflammation.

Rank	Author	Record count	% OF 408	Citations	Average citation/Publication	Total link strength	
1	He, Chuan	7	1.72%	499	71.29	3	
2	Wang, Wei	7	1.72%	159	22.71	45	
3	Xu, Qiong	7	1.72%	590	84.29	14	
4	Zhang, Lei	5	1.23%	254	50.80	4	
5	Zhong, Xiang	5	1.23%	188	37.60	12	
6	Chen, Xi	4	0.98%	23	5.75	23	
7	Chen, Yingyu	4	0.98%	18	4.50	25	
8	Chen, Zheng	4	0.98%	141	35.25	6	
9	Gao, Xiaoli	4	0.98%	15	3.75	45	
10	Gun, Shuangbao	4	0.98%	15	3.75	45	

Table 4 Top 10 co-cited authors in m6A and inflammation.

Rank	Co-cited authors	Citations	Total link strength	
1	Xiang Wang	266	4,875	
2	Kate D Meyer	165	3,383	
3	Dan Dominissini	125	2,572	
4	Hailing Shi	113	2,326	
5	Huilin Huang	108	2,521	
6	Guifang Jia	104	2,060	
7	Yun Yang	102	1,710	
8	Ying Wang	99	1,889	
9	Ian A Roundtree	97	2,087	
10	Jiao Liu	90	1,876	

Figure 4 The visualization of authors (A) and co-cited authors (B).

The size of nodes and edges is weighted by the number of published articles. The colors of the nodes represent different clusters.

Bibliometric analysis of journals and co-cited journals

A total of 225 journals published the documents. As illustrated in Table 5, Frontiers in Immunology emerged as the leading journal with the highest volume of articles pertaining to m6A and inflammation, featuring 18 published papers. This was followed by Frontiers in Genetics (n = 15), Frontiers in Cell and Developmental Biology (n = 14), International Journal of molecular sciences (n = 10), Nature communications (n = 8), Frontiers in pharmacology (n = 7). The citation network of journals is depicted in Fig. 5A. Table 6 shows the top 20 journals with the most significant of co-citations, with the top 5 being Nature (n = 1,040), Cell (n = 728), Nucleic Acids Research (n = 578), Molecular Cancer (n = 559) and Molecular Cell (n = 544). The co-citation journals network is illustrated in Fig. 5B.

Table 5 Top 20 journals in m6A and inflammation.

Rank	Source	Record count	JCR	IF	
1	Frontiers in Immunology	18	Q2	5.7	
2	Frontiers in Genetics	15	Q3	2.8	
3	Frontiers in Cell and Developmental Biology	14	Q2	4.6	
4	International Journal of Molecular Sciences	10	Q2	4.9	
5	Nature Communications	8	Q2	14.7	
6	Frontiers in Pharmacology	7	Q1	4.4	
7	Aging-US	6	Q2	3.9	
8	Bioengineered	6	Q2	4.2	
9	Cell Death & Disease	6	Q2	8.1	
10	Journal of Cellular and Molecular Medicine	6	Q2	4.3	
11	Biochemical and Biophysical Research Communications	5	Q3	2.5	
12	Cellular Signalling	5	Q3	4.4	
13	International Immunopharmacology	5	Q2	4.8	
14	Frontiers in Cardiovascular Medicine	4	Q2	2.8	
15	Molecular Cancer	4	Q1	27.7	
16	Oxidative Medicine and Cellular Longevity	4	Q2	7.31	
17	Aging and Disease	3	Q2	7	
18	Animals	3	Q3	2.7	
19	Biomedicine Pharmacotherapy	3	Q2	6.9	
20	Cell Reports	3	Q1	7.5	

Figure 5 The visualization of journals by citations (A); the co-citation network of journals (B).

Node and edge sizes are weighted by the number of published articles. The colors of nodes indicate different clusters.

Table 6 Top 20 co-cited journals in m6A and inflammation.

Rank	Co-cited journal	Frequency	JCR	IF	
1	Nature	1,040	Q1	50.5	
2	Cell	728	Q1	45.5	
3	Nucleic Acids Research	578	Q1	16.6	
4	Molecular Cancer	559	Q1	27.7	
5	Molecular Cell	544	Q1	14.5	
6	Nature Communications	517	Q1	14.7	
7	Cell Research	421	Q1	28.1	
8	Proceedings of the National Academy of Sciences of the United States of America	353	Q1	9.4	
9	Science	325	Q1	44.7	
10	Frontiers in Immunology	287	Q2	5.7	
11	International Journal of Molecular Sciences	286	Q2	4.9	
12	Cell Reports	257	Q1	7.5	
13	Cell Death & Disease	245	Q2	8.1	
14	Nature Chemical Biology	226	Q1	12.9	
15	The Journal of Biological Chemistry	221	Q2	4	
16	Nature Immunology	208	Q1	27.7	
17	Nature Reviews Molecular Cell Biology	206	Q1	81.3	
18	Frontiers in Cell and Developmental Biology	199	Q2	4.6	
19	Biochemical and Biophysical Research Communications	190	Q3	2.5	
20	Cancer Cell	186	Q1	48.8	

Bibliometric analysis of co-cited references

The 10 most frequently referenced sources are presented in Table 7. The most cited reference was a dissertation authored by Professor Wang et al. (2014) (n = 111), followed by (n = 103) Dominissini et al. (2012), and Wang et al. (2015) (n = 89). Among the top 10 most cited publications, nine were original research articles and one was a review article.

Table 7 Top 10 co-cited references in m6A and inflammation.

Rank	Cited reference	Citations	Total link strength	
1	Wang X, 2014, nature, v505, p117, doi 10.1038/nature12730	111	1,250	
2	Dominissini D, 2012, nature, v485, p201, doi 10.1038/nature11112	103	1,355	
3	Wang X, 2015, cell, v161, p1388, doi 10.1016/j.cell.2015.05.014	89	1,169	
4	Jia GF, 2011, nat chem biol, v7, p885, doi 10.1038/nchembio.687	88	1,218	
5	Liu JZ, 2014, nat chem biol, v10, p93, doi 10.1038/nchembio.1432	82	1,157	
6	Zheng GQ, 2013, mol cell, v49, p18, doi 10.1016/j.molcel.2012.10.015	76	1,148	
7	Meyer KD, 2012, cell, v149, p1635, doi 10.1016/j.cell.2012.05.003	72	899	
8	Li HB, 2017, nature, v548, p338, doi 10.1038/nature23450	61	799	
9	Zaccara S, 2019, nat rev mol cell bio, v20, p608, doi 10.1038/s41580-019-0168-5	61	629	
10	Huang HL, 2018, nat cell biol, v20, p285, doi 10.1038/s41556-018-0045-z	59	851	

Bibliometric analysis of keywords

Initially, we examined the frequency and co-occurrence of keywords utilizing VoSviewer. The top 20 keywords are presented in Table 8. The most prevalent keyword identified was inflammation (n = 127), followed by messenger-RNA (n = 99), N6-methyladenosine (n = 95), methylation (n = 91), and expression (n = 83). The keyword co-occurrence network is illustrated in Fig. 6A. Subsequently, we performed a cluster analysis of the keywords (Fig. 6B). Modularity q-value = 0.5046, Silhouette s-value = 0.7688. The results showed that all the keywords could be categorized into 12 distinct groups, which include FOXO3, m6A modification, immune system, immune infiltration, metabolism, activation, IPEC-J2, DNA methylation, presynaptic, ischemia-reperfusion (i/r), polymyositis and agonist. Additionally, we created a timeline viewer of the keywords by CiteSpace (Fig. 7A). Furthermore, we aggregated all the data from 1996 to 2023 to identify burst keywords through CiteSpace. Burst keywords are defined as those that exhibit a significant frequency of occurrence within a specific timeframe. This analysis not only highlighs the evolution of research hotspots over time but also reflects recent research trends and may forecast future directions. Figure 7B shows the top 13 keywords with the highest incidence of outbreaks. The prominent terms that emerged in the last 5 years include “promotes”, “microRNAs”, “METTL3”, “hepatocellular carcinoma”, and “obesity”. However, the intensity of these burst keywords remains below 2. We interpret this as indicative of the field’s nascent status, with a substantial volume of literature primarily published post-2019. Notably, the role of m6A in inflammatory regulation has garnered significant scholarly interest since 2019 and is expected to continue to be a focal point in future research.

Table 8 The top 20 keywords in m6A and inflammation.

Rank	Keyword	Counts	Total link strength	
1	Inflammation	127	686	
2	Messenger-RNA	99	639	
3	N6-methyladenosine	95	516	
4	Methylation	91	524	
5	Expression	83	430	
6	N-6-methyladenosine	74	422	
7	m(6)A	72	432	
8	METTL3	56	306	
9	Nuclear-RNA	55	356	
10	Cells	50	270	
11	M6a	48	275	
12	Gene-expression	45	265	
13	Translation	45	319	
14	RNA	38	179	
15	RNA methylation	37	229	
16	Cancer	36	212	
17	FTO	29	176	
18	Activation	28	137	
19	Proliferation	27	176	
20	Mechanisms	26	162	

Figure 6 The keyword co-occurrence network (A); the keywords cluster analysis (B).

The size of nodes and edges is weighted by the number of published articles. The colors of the nodes represent different clusters. All of the keywords could be classified into 12 categories, which were FOXO3, m6A modification, immune system, immune infiltration, metabolism, activation, IPEC-J2, DNA methylation, presynaptic, ischemia-reperfusion (i/r), polymyositis and agonist.

Figure 7 The timeline viewer of keywords (A). Top 13 keywords with the strongest citation bursts (B).

The years between “beginning” and “end” represent periods when keywords were more influential. Years in light green indicate that the keyword has not yet appeared, years in dark green indicate that the keyword has less influence, and years in red indicate that the keyword has more influence.

Database cross-validation

To substantiate our findings, bibliometric analyses were were performed utilizing the PubMed and Scopus databases. A total of 372 publications were extracted from the PubMed database, while 581 publications were sourced from Scopus. In the PubMed database, publications released after 2019 represented a substantial majority, comprising 97.84% (364/372) of all entries. The authors with the highest publication counts were Chuan He (n = 8) and Wei Wang (n = 7). The most prevalent keywords in this dataset were “humans”, “animals”, “inflammation”, “mice”, and “adenosine.” The Scopus database provides a more comprehensive compilation of publications, with works published post-2019 accounting for 91.39% (531/581) of the total. The countries with the most significant contributions to these publications were China (n = 477), the United States (n = 66), Germany (n = 17), the United Kingdom (n = 14), and Japan (n = 12). Notably, journals with the highest publication counts comprised Frontiers in Immunology (n = 30), Frontiers in Genetics (n = 19), Frontiers in Cell and Developmental Biology (n = 14), International Journal of Molecular Sciences (n = 12), and Aging (n = 9). The most common keywords within this context included “human”, “article”, “6 N methyladenosine”, “controlled study”, and “nonhuman”. It is crucial to acknowledge that bibliometric analyses concerning institutions may not be directly comparable due to inherent discrepancies in data characteristics between PubMed and Scopus. Despite some observed variations across different databases, overarching trends regarding annual publication rates, participating countries/regions, leading authors, prominent journals, and recurring keywords displayed analogous patterns. Detailed results from these bibliometric analyses are accessible in the Supplemental Materials.

The mechanism of the m6A modification involvement in the inflammatory response

As the most widespread internal mRNA modification, m6A plays a significant role in the inflammatory response to disease. Some mechanisms are summarized and elucidated by bibliometric analysis and systematic organization of keywords. The analysis highlighted that the key mechanisms through which m6A regulates inflammation encompass apoptosis, autophagy, oxidative stress, immune cell dysfunction, and dysregulation of signaling pathways.

m6A and apoptosis

Apoptosis is the first recognized form of programmed cell death, typically occurring during developmental processes and aging, and it functions as a homeostatic mechanism to maintain cellular populations within tissues (Elmore, 2007). The apoptotic mechanism is intricate, with current studies indicating the existence of two primary apoptotic pathways: the extrinsic and intrinsic pathways. Notably, there is an intersection between the two pathways. In addition, there exists an additional pathway involving T cell-mediated cytotoxicity and perforin-granzyme-dependent cell killing. The extrinsic, intrinsic, and perforin/granzyme pathways converge on a common execution pathway (Xu, Lai & Hua, 2019; Lopez & Tait, 2015; Zhang & Xia, 2023; Kashyap, Garg & Goel, 2021). During the process of apoptosis, several proteins are crucial in modulating the apoptotic response, including both anti-apoptotic proteins and pro-apoptotic proteins. Anti-apoptotic proteins include B-cell lymphoma 2 (Bcl-2) and B-cell lymphoma-extra large (Bcl-xl), while pro-apoptotic proteins include Bcl-2-associated X protein (Bax) and Bcl-2 antagonist/killer (Bak) (Zhang & Xia, 2023). Dysregulation of apoptosis has been implicated in various inflammatory diseases (Vallejo-Garcia et al., 2012; Littlewood & Bennett, 2003; Zhan et al., 2024). Furthermore, m6A modifications influence apoptosis by directly or indirectly modulating apoptosis-related factors or pathways (Xu et al., 2021). Figure 8 illustrates the primary mechanisms by which m6A regulates apoptosis.

Figure 8 Effects of m6A on apoptosis.

Apoptosis includes extrinsic, intrinsic, and perforin/granzyme pathways mediated by T cells, all three of which ultimately intersect on the same executive pathway. METTL3 and FTO directly regulate the expression of pro-apoptotic proteins BAK and BAX, and anti-apoptotic proteins BCL-2 and BCL-XL, as well as caspases3, etc. METTL3, FTO, and YTHDF3, IGF2BP2 can also affect apoptosis by regulating the expression of certain genes. Created with BioRender.com.

METTL3 serves different functions in apoptosis through various mechanisms. In the prostate cancer cell lines LNCaP and PC3, the depletion of METTL3 results in increased expression levels of the pro-apoptotic proteins Bak and Bax, alongside heightened activity of caspase3 and caspase7, while simultaneously decreasing the expression levels of the anti-apoptotic proteins Bcl-2 and Bcl-xl (Cai et al., 2019). This study does not address the m6A modification levels in mRNAs of apoptosis-related proteins; thus, investigating the m6A modification levels of target genes may provide deeper insights into the mechanistic role of METTL3. In human myeloid leukemia (MOLM13) cells with METTL3 depletion, the level of m6A modification of BCL-2 is reduced, correlating with decreased protein expression. The upregulation of METTL3 inhibits apoptosis by enhancing the translation of MYC, BCL2, and PTEN mRNAs through increased methylation (Vu et al., 2017). The expression of METTL3 was diminished in both in vivo temporomandibular joint osteoarthritis (TMJ OA) mouse models and in vitro TNF-α-stimulated mouse chondrocytes (ATDC5). MeRIP analysis indicated that METTL3 overexpression significantly increased the level of m6A methylation of Bcl-2 mRNA. The m6A modification installed by METTL3 on target mRNAs was dependent on reader recognition. Further experiments showed that YTHDF1 mediated Bcl-2 mRNA stability in an m6A-dependent manner. In conclusion, overexpression of METTL3 increases Bcl-2 mRNA stability through YTHDF1-mediated m6A modification, thereby inhibiting TNF-α-induced apoptosis and autophagy in ATDC5 (He et al., 2022). Furthermore, METTL3 overexpression increased the m6A modification level of Bax mRNA in a cardiomyocyte cell line (HL1) following in vitro hypoxia/reperfusion injury (iH/R), potentially safeguarding cardiomyocytes from iH/R-induced apoptosis by downregulating pro-apoptotic gene expression (Su et al., 2021). Conversely, another study indicated that METTL3 had an opposing effect on apoptosis. METTL3 knockdown significantly mitigated inflammation and apoptosis in renal podocytes from diabetic patients, and its overexpression significantly exacerbated these responses in vitro. Further studies revealed that METTL3 regulates Notch signaling and exerts pro-inflammatory and pro-apoptotic effects through the m6A modification of TIMP2 in an insulin-like growth factor 2 mRNA binding protein 2 (IGF2BP2)-dependent manner (Jiang et al., 2022).

Moreover, significantly increased expression levels of METTL14 were observed in the kidneys of mice suffering from diabetic nephropathy. METTL14-dependent RNA m6A modification contributes to podocyte damage by regulating Sirt1 mRNA post-transcriptionally. The knockdown of METTL14 resulted in the inhibition of apoptosis and inflammation in podocytes (Lu et al., 2021).

In myocardial ischemia-reperfusion injury (IRI) mice and hypoxia/reoxygenation (H/R)-induced cardiomyocytes (CMs), the m6A eraser fat mass and obesity-associated protein (FTO) is downregulated. Further studies indicate that FTO target Yes-associated protein (YAP1) mRNA in CMs. As a transcriptional co-activator, YAP1 was shown to prevent CM apoptosis by promoting AKT signaling (Del Re et al., 2013). Overexpression of FTO enhances the stability of YAP1 mRNA by demethylating it, thereby attenuating H/R-induced apoptosis and inflammation in CMs (Ke et al., 2022). Furthermore, FTO overexpression reduces cleaved caspase-3 and Bax levels and increased Bcl-2 expression (Shen et al., 2021). Silencing of ALKBH5 inhibits apoptosis and suppresses ultraviolet B (UVB)-induced cellular pyroptosis and inflammatory responses in patients with chronic actinic dermatitis (CAD) (He et al., 2023b).

The m6A reader YTHDF3 binds to the 3′UTR m6A modification site of mitochondrial calcium unidirectional transport protein (MCU), a key protein in human cytomegalovirus (HCMV)-induced apoptosis in vascular endothelial cells, enhancing MCU translation and promoting apoptosis (Zhu, Zhang & Wang, 2022). YTHDF3 or IGF2BP2 knockdown suppresses H/R-induced apoptosis in BEAS-2B cells by deactivating the p38, ERK1/2, AKT, and NF-κB pathways (Xiao et al., 2022).

m6A and autophagy

Autophagy is a highly conserved process of eukaryotic cell death and recycling. Through the degradation of cytoplasmic organelles, proteins, and macromolecules, and the recycling of catabolic products, autophagy plays a critical role in cell survival and maintenance. Numerous autophagy-related (ATG) genes regulate this process (Parzych & Klionsky, 2014). Disruption of autophagy affects various inflammatory conditions, including infectious and non-infectious diseases such as autoimmune disorders, tumors, and neurodegenerative diseases (Deretic, 2021). The m6A modification may influence the transcriptional regulation of ATG genes, thereby affecting autophagy. Figure 9 illustrates the primary mechanisms by which m6A modulates autophagy.

Figure 9 The main mechanism by which m6A regulates autophagy.

METTL3 promotes autophagy by promoting the expression of autophagy-associated proteins such as ATG5 and ATG7. On the contrary, METTL3 can also inhibit autophagy by increasing the expression of Foxo3a. In addition, METTL3 decreases the expression level of TFEB, which inhibits lysosomal synthesis and reduces autophagy. ALKBH5 promotes autophagy by promoting TFEB expression. FTO promotes autophagy initiation by specifically upregulating ULK1, a key protein kinase in the regulation of autophagy initiation. YTHDC1 increases the stability of the autophagy receptor SQSTM1 nuclear mRNA thereby regulating autophagy. Created with BioRender.com.

The effects of m6A modifications on autophagy may are primarily inhibitory, with METTL3 shown to negatively regulate autophagy in a variety of contexts. In H/R-induced cardiomyocytes, METTL3 methylates transcription factor EB (TFEB), a key regulator of lysosomal biosynthesis and autophagy genes, at the 3′-UTR. The m6A modification promotes the binding of the RNA-binding protein HNRNPD to the TFEB precursor mRNA, subsequently promoting the degradation of the TFEB mRNA, which results in decreased expression levels and reduced autophagy (Song et al., 2019). In primary hepatocellular carcinoma (HCC), METTL3 adds m6A modification to the 3′-UTR of FOXO3 mRNA, a negative regulator of autophagy, increasing its mRNA stability in a YTHDF1-dependent manner (Lin et al., 2020). Conversely, some studies report that METTL3 overexpression increased m6A levels on ATG5 and ATG7 transcripts, promoting their expression and enhancing autophagy (Chen et al., 2021; Liu et al., 2020a).

The m6A eraser FTO is also a crucial regulator of autophagy, functioning as a positive regulator. FTO promotes autophagy initiation by specifically upregulating the protein levels of Unc-51-like kinase 1(ULK1), a key kinase in autophagy initiation (Jin et al., 2018). FTO silencing leads to higher m6A levels on ATG5 and ATG7 transcripts, causing their degradation and reduced protein expression, thereby inhibiting autophagy (Wang et al., 2020). ALKBH5 similarly promotes autophagy by demethylating TFEB mRNA (Song et al., 2019).

The m6A reader YTHDC1 regulates autophagy by stabilizing the nuclear mRNA of the autophagy receptor SQSTM1. YTHDC1 knockdown leads to SQSTM1 mRNA degradation, reducing autophagic flux (Liang et al., 2022). YTHDF1 enhances the translation of ATG2A and ATG14 in an m6A-dependent manner, regulating hypoxia-induced autophagy and autophagy-related HCC progression (Li et al., 2021). YTHDF3 promotes autophagy by recognizing the m6A modification site near the FOXO3 mRNA termination codon and facilitates FOXO3 translation by recruiting eIF3a and eIF4B (Hao et al., 2022).

m6A and oxidative stress

Oxidative stress arises from the excessive generation of reactive oxygen species (ROS) in cells or tissues, which disrupts redox homeostasis due to an imbalance between ROS generation and scavenging mechanisms. This disruption may be a significant mechanism in the development of chronic inflammation (Hussain et al., 2016). The m6A modification plays a critical role in regulating cellular responses to oxidative stress. Figure 10 illustrates the primary mechanisms by which m6A participates in the oxidative stress response.

Figure 10 Main mechanisms involved in the regulation of oxidative stress by m6A.

METTL3 increases the activity of intracellular antioxidant enzymes such as SOD, CAT, and GSH-PX to resist oxidative stress by regulating the Keap1-Nrf2 pathway. Overexpression of FTO up-regulates the expression of genes involved in mitochondrial biosynthesis and oxidative phosphorylation, thereby affecting the mitochondrial oxidative respiratory chain and enhancing the level of oxidative stress. YTHDF3 inhibits mitochondrial oxidative stress by regulating PRDX3 in a m6A-dependent manner. manner regulates PRDX3 and inhibits mitochondrial oxidative stress. Meanwhile, oxidative stress induced by hydrogen peroxide and arsenite increased the level of intracellular m6A methyltransferase and thus the overall level of m6A modification. Created with BioRender.com.

Oxidative stress triggered by agents such as hydrogen peroxide and arsenite elevates intracellular levels of m6A methyltransferases (Li et al., 2017). Zhao et al. (2019) demonstrated that arsenite-induced oxidative stress increased the level of m6A modification by promoting WTAP and METTL14 protein expression. Conversely, when arsenite-induced oxidative stress was inhibited by antioxidant N-acetylcysteine (NAC), m6A modification and methylase levels returned to baseline (Zhao et al., 2019; Pedre et al., 2021). Furthermore, METTL3 may serve a protective function in oxidative stress. Wang et al. (2019a) found that overexpression of METTL3 in mouse renal tubular epithelial cells (mRTECs) led to significantly lower ROS levels and lipid peroxidation marker malondialdehyde (MDA), along with increased activity of antioxidant enzymes such as superoxide dismutase (SOD), catalase (CAT), and glutathione peroxidase (GSH-PX). Further investigation revealed that METTL3 interacts with the microprocessor protein DiGeorge syndrome critical region gene 8 (DGCR8), which regulates miRNA biogenesis, and positively regulated the miR-873-5p maturation process in an m6A-dependent manner. miR-873-5p regulated the Keap1-Nrf2 pathway, thereby counteracting fucoxanthin-induced oxidative stress and cellular apoptosis (Wang et al., 2019a).

The m6A eraser FTO is also associated with oxidative stress. A significant increase in palmitate-induced ROS production was observed in human myotubular cells overexpressing FTO (Bravard et al., 2011). Zhuang et al. (2019) observed specific upregulation of genes related to mitochondrial biosynthesis (PGC-1α, NRF1, and TFAM) and oxidative phosphorylation (Cox5a, Atp5 g1, Atp5a1) in FTO overexpressing cells. Mitochondria, as key regulators of apoptosis and oxidative stress in mammalian cells, are affected by FTO overexpression, which alters the mitochondrial respiratory chain and intensifies oxidative stress (Guo et al., 2013). This further elucidates the functional mechanism of FTO in oxidative stress.

m6A readers exhibit diverse functions in oxidative stress. In endothelial cells, reduced expression of YTHDC1 correlates with increased ROS levels and decreased SOD2 expression (Yin et al., 2023). Translation of Peroxiredoxin 3 (PRDX3), a major regulator of mitochondrial oxidative stress, is directly regulated by YTHDF3 in an m6A-dependent manner. Overexpression of PRDX3 prevents mitochondrial oxidative stress-induced injury in a variety of diseases (Arkat et al., 2016; Hwang et al., 2019). PRDX3 expression was suppressed when YTHDF3 was knocked down (Sun et al., 2022).

m6A and immune cell dysregulation

The bibliometric analysis of keywords indicates that m6A is involved in modulating the inflammatory response by regulating the function of innate immune cells such as neutrophils, macrophages, and NK cells, in addition to adaptive immune cells including T cells and B cells (Fig. 11).

Figure 11 M6A is involved in regulating the immune function of innate immune cells neutrophils, macrophages, NK cells and adaptive immune cells T cells and B cells.

Neutrophils: METTL3 regulates neutrophil release from the bone marrow to the circulation in a TLR4 signalling-dependent manner through surface expression of CXCR2. ALKBH5 promotes neutrophil migration to the site of infection by increasing the expression of neutrophil migration-promoting molecules CXCR2 and NLRP12, and by decreasing the expression of neutrophil migration-inhibiting molecules PTGER4, TNC and WNK1. the site of infection. Macrophage: METTL3 promotes M1 macrophage polarisation through STAT1 and HDAC5, etc. FTO promotes macrophage polarisation through the NF-κB signalling pathway. YTHDF2 induces M2 macrophage polarisation by inhibiting the expression of p53 and inhibits M1 macrophage polarisation through inhibition of the NF-κB, p38 and JNK signalling pathways. NK cells: FTO inhibits NK cell activation and function by inhibiting SOCS JAK/STAT signalling. YTHDF2 promotes NK cell effector function through the formation of a positive feedback loop of STAT5-YTHDF2. T Cells: METTL3 promotes the differentiation of naïve T cells to Th1 and Th17 cells and inhibits the differentiation of naive T cells to Th1 and Th17 cells by regulating the activity of the SOCS family and mediating the activation of STAT5. Th17 cells and inhibits differentiation to Th2 cells without affecting Treg. In addition, ALKBH5 promotes Jag1/Notch2 signalling and inhibits the expansion of thymic γδ T cells. B cells: METTL14 mediates pro-B cell proliferation and cell volume increase while regulating the transformation of large pre-B cells to small pre-B cells. In addition, METTL14 has a crucial role in GC B-cell responses. METTL3 mainly affects the ability of GC B cells to mount an effective immune response, whereas it has no significant effect on B-cell development. Created with BioRender.com.

Neutrophil

Neutrophils represent a crucial element of the innate immune response and constitute the predominant leukocyte population in peripheral blood. Upon pathogen invasion, neutrophils are mobilized from the bone marrow into the bloodstream and migrate to the infection site in the early stages (Shen et al., 2017). The migration and aggregation of neutrophils at the site of infection are critical for the effective innate immune. A thorough understanding of the molecular mechanisms governing neutrophil migration is essential for enhancing the body’s anti-inflammatory response while preventing inflammatory tissue damage. Neutrophil migration and aggregation are mediated by extracellular signals such as chemokines and cytokines, as well as chemokine receptors, cytoskeletal proteins, and intracellular signaling pathways (Park et al., 2019). METTL3 mediates m6A modification of Toll-like receptor 4 (TLR4) mRNA to promote its protein expression. In lipopolysaccharide (LPS)-induced endotoxemia, METTL3 controls neutrophil release from the bone marrow into the circulation in a TLR4-signaling-dependent mechanism involving surface expression of CXC chemokine receptor 2 (CXCR2) (Luo et al., 2023). ALKBH5 demethylates neutrophil migration-associated molecules and alters the fate of target RNAs. ALKBH5 increases the expression of the neutrophil migration-promoting molecules CXCR2 and NLR family pyrin domain containing 12 (NLRP12), while reducing the expression of the neutrophil migration-suppressing molecules prostaglandin E receptor 4 (PTGER4), tenascin-C (TNC), and with-no-lysine kinase 1 (WNK1). This intrinsic epigenetic mechanism enables the accumulation of neutrophils at infection sites and, promotes effective bacterial clearance, thereby preventing an excessive inflammatory response (Qian & Cao, 2022).

Macrophage

Macrophages initiate immune response by recognizing pathogen-associated molecular patterns (PAMPs) from invading pathogens and damage-associated molecular patterns (DAMP) from stress or tissue injury. Upon activation by external stimuli, macrophages trigger complex cascade of responses, demonstrating plasticity (M1 and M2 polarization) and pluripotency (pro-inflammatory and anti-inflammatory) (Shapouri-Moghaddam et al., 2018). Pooled CRISPR screening has identified METTL3 as a positive regulator in the innate immune responses of macrophages. METTL3 deficiency in macrophages impairs their resistance to pathogens and tumors. METTL3 mediates the m6A-modified of interleukin 1 receptor-associated kinase 3 (IRAK3) mRNA, accelerating its degradation and thereby promoting macrophage activation. In METTL3-deficient macrophages, elevated IRAK3 expression reduces TLR4 signaling, which negatively regulates macrophage activity (Tong et al., 2021). METTL3 is upregulated in M1-polarized macrophages, and overexpression of METTL3 promotes M1 polarization while inhibiting M2 polarization. The regulation of macrophage polarization by METTL3 involves multiple signaling pathways. Further studies have shown that METTL3 directly catalyzes the m6A modification of STAT1 mRNA, a transcription factor critical for M1 macrophages activation, thereby enhancing mRNA stability and expression (Liu et al., 2019b). However, some studies have found that METTL3 exhibits an inhibitory effect on macrophage inflammation. For instance, METTL3 reduces LPS-induced inflammatory responses in macrophages via the NF-κB signaling pathway (Wang et al., 2019b). Additionally, METTL3 inhibits M1 macrophage polarization by degrading m6A-modified SOCS2 mRNA (Zhong et al., 2021). The differing functions of METTL3 in macrophages may result from variations in the cellular microenvironment. In addition, macrophages deficient in METTL14 or YTHDF1 exhibit impaired SOCS1 induction, leading to increased pro-inflammatory cytokine and chemokine production, which exacerbates responses to bacterial infections (Du et al., 2020). FTO knockdown inhibited NF-κB signaling pathway and decrease the mRNA stability of signal transducer and activator of transcription 1 (STAT1) and peroxisome proliferator-activated receptor-γ(PPAR-γ) through YTHDF2, thereby obstructing macrophage activation (Gu et al., 2020). ALKBH5 promotes the activation of JNK and ERK pathways by upregulating MAP3K8, which regulates IL-8 expression and promotes macrophage recruitment (You et al., 2022). YTHDF2 facilitates M2 macrophage polarization by promoting p53 mRNA degradation and inhibits M1 macrophage polarization by suppressing NF-κB, p38, and JNK signaling pathways (Cai et al., 2022a).

NK cell

Natural killer (NK) cells, a core component of the innate immune system, are granular lymphocytes with inherent cytotoxicity, enabling them to target and kill foreign, transformed, or infected cells (Pfefferle et al., 2020). Recent findings indicate that m6A serves as a positive regulator of NK cell-mediated antitumor immunity. METTL3 expression correlates positively with NK cell effector molecule levels and functional. Research by Song et al. (2021) highlighted a decrease in METTL3 expression within tumor-infiltrating NK cells. The depletion of METTL3 in NK cells disrupts their homeostasis and hampers both their infiltration and function in the tumor microenvironment. The gene encoding src homology-2-containing protein tyrosine phosphatase 2 (SHP-2) undergoes m6A modification, with METTL3-mediated m6A methylation enhancing its expression. In NK cells, SHP-2 facilitates the activation of the AKT-mTOR and MAPK-ERK signaling pathways in response to IL-15 stimulation. METTL3 deficiency thus diminishes NK cell effector function (Song et al., 2021). FTO acts as a negative regulator of IL-2/15-driven JAK/STAT signaling by stabilizing the mRNA of SOCS family suppressor genes, thereby inhibiting NK cell activation and function (Kim et al., 2023). NK cells exhibit higher levels of YTHDF2 compared to other immune cells, and YTHDF2 expression is significantly upregulated in NK cells upon activation by tumors, viral infections, and cytokines, such as IL-15. YTHDF2 maintains NK cell homeostasis, maturation, and IL-15-mediated survival and proliferation, while also promoting NK cell effector functions through the establishment of a STAT5-YTHDF2 positive feedback loop. In contrast, YTHDF2 deficiency impairs NK cells’ antitumor and antiviral activities (Ma et al., 2021b).

T-lymphocyte

T lymphocytes originate from bone marrow progenitors, mature in the thymus, and then become activated, proliferated, and differentiated before being transported to periphery sites for immune function (Du et al., 2020). Double-positive (DP, CD4+CD8+) cells expressing the αβ T-cell antigen receptor in the thymus differentiated into either CD4+T helper cells or CD8+T cytotoxic cells (Collins & Littman, 2005). Under different cytokines stimulations, naïve CD4+T cells may differentiate into diverse subpopulations, including T helper cells (Th1, Th 2, Th 9, Th 17, Th 22, etc.) and T regulatory cells (Treg) (Zhu, Yamane & Paul, 2010). The m6A methylation modification plays a significant role in the differentiation and functionality of CD4+ T cells. Li et al. (2017) demonstrated that deletion of METTL3 in mouse T cells disrupts T cell homeostasis and differentiation. METTL3-deficient T cells remained in their initial state for 12 weeks and failed to undergo stable proliferation. Furthermore, METTL3-deficient naïve T cells exhibited decreased methylation of SOCS family gene transcripts, leading to an increased SOCS family mRNA and protein expression. Elevated SOCS activity suppresses interleukin-7 (IL-7)-mediated STAT5 activation, thereby inhibiting homeostatic T cell proliferation and differentiation. Moreover, METTL3-deficient naïve T cells differentiated into fewer Th1 and Th17 cells and more Th2 cells than wild-type (WT) naïve T cells, without affecting Treg induction (Li et al., 2017). Research also suggests that METTL14 deficiency affects invariant NKT (iNKT) development through cell-intrinsic mechanisms, leading to impaired iNKT cell maturation and differentiation. Mechanistically, the disruption of METTL14-dependent m6A modification alters expression of key molecules involved in the p53-mediated apoptotic pathway, potentially increasing apoptosis in DP thymocytes and iNKT cells. In addition, METTL14 knockdown in mature iNKT cells results in increased Cish expression and diminished TCR signaling, which in turn reduces their cytokine production (Cao et al., 2022). Ding et al. (2022) discovered that ALKBH5 depletion in lymphocytes decreased Jag1/Notch2 signaling by increasing m6A RNA modification in thymocytes, which specifically induces γδ T cell expansion (Zhao, Ding & Li, 2023).

B-lymphocyte

B-lymphocytes play a central role in adaptive humoral immunity by producing antigen-specific immunoglobulins (Ig) that target invasive microorganisms (Vaughan, Roghanian & Cragg, 2011). In humans, B cells also originate from bone marrow hematopoietic stem cells. Hematopoietic stem cells differentiate into multipotent progenitor cells (MPPs), which lack self-renewal capacity, and lymphoid-induced multipotent progenitor cells (LMPPs), which can give rise to common lymphoid progenitor cells (CLPs). The B cell-biased lymphoid progenitor cells (BLPs) within the CLPs will further develop into B cells. The B-cell development process follows an intrinsic program, passing through several intermediary stages, including progenitor cells (Pro-B cells) and precursor B cells (large Pre-B cells, Small Pre-B cells), before entering the peripheral blood. Subsequently, immature B cells migrate to the spleen for further maturation (Eibel et al., 2014; Rolink et al., 1999). The m6A modification is crucial in regulating early B cell proliferation and development (Zhao et al., 2022). The deletion of both METTL14 and YTHDF2 in B cells significantly blocked IL-7-induced Pro-B cell proliferation and development. Loss of METTL14 hinders Pro-B cell proliferation and cell growth in vitro and leads to a reduced viability of large Pre-B cells in vivo. In addition, METTL14 deficiency disrupts IL-7-driven transition from Pro-B cell to Small Pre-B cell, resulting in aberrant gene expression patterns crucial for B cell development and significantly impeding early B cell maturation.YTHDF2-mediated mRNA degradation is also critical for the transition from Pro-B to small Pre-B, with METTL14 deletion in B cells reducing YTHDF2 binding to its target mRNAs, thereby affecting early development (Zheng et al., 2020). METTL14 also plays a vital role in germinal center (GC) B cell responses. In a YTHDF2-dependent manner, METTL14-mediated m6A modification positively regulates GC B cell responses. Mechanistically, METTL14-mediated m6A promotes mRNA decay of genes encoding negative immunomodulatory factors, such as Lax1 and Tipe2, thereby indirectly upregulating genes essential for positive selection and proliferation of GC B cells. METTL14 deficiency results in elevated Myc proteins levels in GC B cells and vitro-stimulated B cells (Huang et al., 2022). By contrast, Grenov et al. (2021) found that no significant alterations in B cell development in the METTL3-deficient mouse model. However, METTL3-deficient GC B cells exhibited reduced cell cycle progression and lower expression of genes related to proliferation and oxidative phosphorylation, resulting in a marked reduction in B cell capacity to mount an effective immune response (Grenov et al., 2021).

m6A and dysregulation of inflammatory signaling pathways

Keyword analysis indicates that m6A primarily influences the inflammatory response by regulating the NF-κB pathway, a classic pro-inflammatory signaling pathway. Pro-inflammatory cytokines, such as interleukin-1 (IL-1) and tumor necrosis factor-α (TNF-α), activate NF-κB, which in turn drives the expression of additional pro-inflammatory genes, including cytokines, chemokines, and adhesion molecules (Lawrence, 2009). In the chondrogenic cell line ATDC5, IL-1β treatment increased both overall m6A methylation levels and METTL3 mRNA expression. Silencing METTL3 reduced IL-1β-induced apoptosis and suppressed IL-1β-mediated increases in inflammatory cytokines and NF-κB activation in chondrocytes (Liu et al., 2019a). METTL3 overexpression promotes TRAF6-NF-κB pathway activation in an m6A-dependent manner, enhancing LPS-induced microglial cell inflammation (Wen et al., 2022). METTL14 regulates IL-6 transcription through the NF-κB/p65 pathway. METTL14 knockout mice reduced atherosclerotic plaque formation and inflammatory responses (Zheng et al., 2022a). FTO was reduced in LPS-induced human normal chondrocyte (C28/I2) cells. Overexpression of FTO attenuated LPS-induced C28/I2 cell injury. miR-515-5p is involved in IL-1β-induced chondrocyte apoptosis, inflammation, and extracellular matrix degradation, highlighting its critical role in OA progression (Wu et al., 2021). Further studies revealed that FTO interacts with DGCR8 and regulates pri-miR-515-5p processing in an m6A-dependent manner, and miR-515-5p inactivates the MyD88/NF-κB pathway via TLR4 targeting. Consequently, FTO overexpression suppressed synovial inflammation and alleviated osteoarthritis by inhibiting the miR-515-5p/TLR4/MyD88/NF-κB axis in an m6A-dependent manner (Cai et al., 2023b). Single-cell sequencing (scRNA-seq) and validation studies showed significantly reduced FTO expression in retinal microglia from uveitis mice and in microglia clone 3 (HMC3) cells from inflammatory patients. Glypican 4 (GPC4), a glycosylphosphatidylinositol (GPI)-anchored heparan sulfate proteoglycan (HSPG), serves as a target of FTO and modulates the TLR4/NF-κB pathway by anchoring CD14 (He et al., 2023a).

Discussion

In this study, we performed a bibliometric analysis of research on the involvement of m6A in inflammatory responses to assess the current status and emerging trends in this field. Additionally, we summarized the possible mechanisms through which m6A regulates inflammation based on keyword analysis. These results may provide important insights for future research.

The bibliometric results revealed that studies on m6A and inflammation received limited attention prior to 2018. However, since 2019, there has been exponential growth in the number of annual publications and citations. Publications from 2019–2023 accounted for the vast majority (97.9%) of all publications. This surge in publications is closely related to the discovery of m6A demethylases and the advancements in high-throughput sequencing technology. In 2011, Professor Chuan He first identified FTO as an m6A demethylase, this discovery brought m6A research into a new era of development (Jia et al., 2011). Prof He’s contributions will be discussed in more detail later. The identification of m6A modifications at the genome-wide or transcriptome level remained largely unexplored until 2012, when Meyer and Dominissini first characterized m6A methylation on a large scale and in a high-throughput manner in both human and mouse. This approach, known as Methylated RNA Immunoprecipitation (MeRIP-seq or m6A-Seq), revolutionized the study of RNA modifications (Dominissini et al., 2012; Meyer et al., 2012). In 2015, Linder et al. (2015) developed the m6A Individual Nucleotide Resolution Crosslinking and Immunoprecipitation’ (miCLIP-seq), which enables precise identification of m6A residues at the single-base level, thereby allowing accurate mapping of m6A positions in the transcriptome. These groundbreaking advances, coupled with developments in sequencing technologies, have propelled m6A research into a period of rapid growth. As a result, studies on m6A methylation in inflammation have emerged as a dynamic and highly active area of research in recent years, with promising future directions.

Both China and the United States have been key contributors to the study of m6A involvement in the regulation of inflammation. China stands out as the leading country in terms of publications, citations, and the highest H-index, indicating the high quality of its research output. Chinese research institutions have emerged as dominant players in this field. Among the top 10 institutions, all of which are located in China, are Sun Yat-sen University, Central South University, and Shanghai Jiao Tong University, highlighting the growing attention that m6A research has received in Chinese academic circles, particularly in the context of inflammatory diseases.

Professor He (2010) from the University of Chicago is one of the most prolific authors in the field and a leading figure in m6A methylation research. In 2010, he was the first to conceptualize RNA epigenetics. The following year, Prof. He and his team identified FTO as the first m6A demethylase, demonstrating that m6A modification is reversible (Jia et al., 2011). This groundbreaking discovery marked a significant turning point in the understanding of m6A methylation, revitalizing a field that had remained relatively stagnant for over three decades and propelling it into a dynamic new area of study. Over the next decade, Prof. He’s term continued to contribute to RNA epigenetics. In 2012, they identified another m6A demethylase, ALKBH5 (Zheng et al., 2013). In 2013, they uncovered two critical components of the m6A methyltransferase complex, METTL14 and WTAP. These two proteins, along with METTL3, form a functional complex that catalyzes m6A modification (Liu et al., 2014). The contributions of Prof. He and his team have significantly propelled the field of RNA epigenetics forward. Another prominent researcher in this area is Dr. Qiong Xu, whose publications have significantly advanced our understanding of m6A’s role in inflammation. Dr. Xu’s work has explored how the m6A writer METTL3 and the reader YTHDF2 regulate inflammatory responses by modulating the stability of Pyk2 mRNA and influencing Smad and MAPK signaling pathways (Cai et al., 2023a, 2022b; Yu et al., 2019). In addition, Luo, Xu & Sun (2021) summarized the possible mechanisms by which m6A modifications contribute to various inflammatory states, including autoimmunity, infection, metabolic diseases, and cancer (Luo, Xu & Sun, 2021; Xu et al., 2022; Bechara & Gaffen, 2021). Their work has been crucial in providing a more comprehensive understanding of m6A’s impact on inflammatory diseases. These researchers, through their multifaceted exploration of the mechanisms underlying m6A involvement in inflammation, have made substantial contributions that will guide future research in this rapidly evolving field.

There is typically a strong correlation between the impact of a journal and the influence of the articles it publishes (Callaham, Wears & Weber, 2002). Among the top 20 journals in terms of the number of papers, Molecular cancer had the highest (IF = 37.3), followed by Nature communications (IF = 16.6) and Cell death & disease (IF = 9). A total of 14 journals had an IF greater than 5, indicating the high quality of articles in the field. The Impact factor and Journal Citation Reports (JCR) are widely used metrics for assessing journal impact. JCR categorizes journals into four quartiles (Q1–Q4) based on their IF. Among the top 20 journals by article count, 15% are in Q1 and 65% are in Q2. Furthermore, most of the active journals belong to specialized categories. Some multidisciplinary journals, such as Cell Death & disease and the International Journal of Molecular Sciences, also publish high-quality studies investigating m6A modifications in inflammation, which is an emerging and evolving research topic involving epigenetics, molecular biology, cell biology, biochemistry, and medicine.

Based on keyword mapping, apoptosis, autophagy, oxidative stress, immune cell dysfunction, and dysregulation of signaling pathways appear to be the primary mechanisms by which m6A regulates inflammation. However, m6A modifications exhibit contradictory roles in processes such as apoptosis, autophagy, and other processes. According to existing studies, the effects of m6A on target genes regulation and inflammation progression depend primarily on three factors: whether the target genes modified by m6A promote or inhibit apoptosis or autophagy; the fate of mRNAs after m6A modification; and the readers-dependent regulation of mRNAs post-m6A modification. In addition, several studies have suggested that certain functions of METTL3 and FTO may be independent of m6A modifications, indicating the need for further investigation (Liu et al., 2020b; Zheng et al., 2022b).

In summary, m6A modifications play a broad and significant role in inflammatory regulation through various mechanisms. Given the crucial role of m6A in disease pathways, its potential as a clinical therapeutic target is gaining increasing attention. Small-molecule inhibitors or agonists that target dysregulated m6A regulatory proteins hold promise as potential treatments. For example, STM2457, a potent and selective first-in-class catalytic inhibitor of METTL3, is the most advanced METTL3 inhibitor discovered to date. STM2457 treatment slows the growth of acute myeloid leukemia (AML), while promoting differentiation and apoptosis (Yankova et al., 2021). Additionally, several natural products, including quercetin, baicalein, and lignans, have also been identified as METTL3 inhibitors by virtual screening (Du et al., 2022). Meclofenamic acid (MA), a nonsteroidal anti-inflammatory drug (NSAID), has also been found to act as a highly selective FTO inhibitor, though the potential connection between its anti-inflammatory effects and its role as an FTO inhibitor requires further exploration (Huang et al., 2015). Recently developed FTO inhibitors, such as FB23 and FB23-2, directly bind to FTO, selectively inhibiting its activity. These inhibitors have been shown to selectively inhibit AML cell proliferation and promote differentiation and apoptosis in vitro, significantly inhibiting AML progression in xenograft mice (Huang et al., 2019). While current studies demonstrate the therapeutic potential of targeting m6A regulatory proteins, caution is needed. Due to different cellular environments, m6A modifications in different genes or different regions of the same gene may regulate the fate of mRNAs through different mechanisms, which contributes to the paradoxical roles of m6A in disease pathogenesis. Further studies investigation the regulation of m6A modifications across different transcriptomic regions are essential for a deeper understanding in pathophysiological processes. Therefore, the development of safer and more effective small-molecule inhibitors or agonists targeting m6A-regulated proteins will facilitate the development of RNA-based precision therapies in the future.

Recently, bibliometric studies in related fields have been published, including m6A in viral infections and m6A-related bibliometric studies in cancer. We compared these related studies. Cancer-related m6A studies included a total of 890 publications, much more than the 409 for inflammation and 319 for viral infections. This indicates that m6A methylation modifications have received greater attention in cancer. Similarly, across studies, a large number of publications were focused on 2019 onwards. It indicates that m6A modification is still an emerging field. The reasons for this change in the number of publications have been discussed. Keyword analysis of these related fields revealed that keywords such as nuclear RNA, methylation, messenger RNA, expression, and Cancer were more common keywords in different studies. This may be a common focus of attention in different studies. METTL3 and FTO are the m6A-related proteins that have received the most widespread attention. In the field of Cancer, immunotherapy and prognosis are important areas of research. Researchers in the field of inflammation should pay more attention to the study of m6A modification-related therapies for clinical translation. Through this cross-comparison, the positioning of m6A research in the wider scientific field is revealed. m6A research has a unique position and potential for development in the life sciences, which provides a direction for future research.

It is worth noting that our study suffers from some limitations. First, only the WOSCC was searched in this study, excluding all other databases, such as PubMed and Google Scholar. Papers collected from the WOSCC may have some delays, resulting in some bias in citations and H-index in the study. Also, some key studies will inevitably be missed. However, WOSCC is regarded as a reliable, specialized, and internationally influential citation database, lending credibility to our data selectiont (Liu et al., 2022b). Second, the deadline for the publication collection is 31 December, 2023, meaning that some publications published after this date were not included. Finally, bibliometric analysis dose not account for the scientific rigor of individual publications. The citation count of a publication may not necessarily reflect its scientific quality. Despite these limitations, our study conducted a thorough analysis of the relevant literature, helping researchers better understand the molecular mechanisms of m6A in inflammation regulation and guiding future research directions.

Conclusions

This represents the inaugural comprehensive bibliometric assessment of literature concerning m6A methylation in the context of inflammation. The domain is currently witnessing significant expansion. The results address a notable gap in existing research, aiding scholars in evaluating the characteristics, contribution distribution, and knowledge landscape of m6A methylation within the context of inflammation. These insights may encourage further research in this emerging area. Based on the keyword analysis, we have delineated the primary mechanisms by which m6A regulates inflammation, including apoptosis, autophagy, oxidative stress, immune cell dysfunction, and the dysregulation of inflammation-related signaling pathways. The findings highlight key research trends and areas of interest, enhancing our comprehension of the current research landscape and offering important guidance for future investigations and clinical applications.

Supplemental Information

Supplemental Information 1 Trends in growth of publications from PubMed (A) and Scopus (B) databases.

Supplemental Information 2 The visualization of institutions (A), authors(B) and keywards(C) from PubMed.

Node and edge sizes are weighted by the number of published articles. The colors of nodes indicate different clusters.

Supplemental Information 3 The visualization of countries regions (A), institutions (B), journals (C) and authors (D) from Scopus.

The size of nodes is weighted by the number of published articles.

Supplemental Information 4 The visualization of co-cited authors (A) and co-cited journals (B) from Scopus.

Node and edge sizes are weighted by the number of published articles. The colors of nodes indicate different clusters.

Supplemental Information 5 The keyword co-occurrence network (A), top 16 keywords with the strongest citation bursts (B), the keywords Cluster Analysis (C) and the timeline viewer of keywords (D) from Scopus.

The keyword co-occurrence network (A). The size of nodes and edges is weighted by the number of published articles. The colors of the nodes represent different clusters. The size of nodes and edges is weighted by the number of published articles. The colors of the nodes represent different clusters. Top 16 keywords with the strongest citation bursts (B). The years between “beginning” and “end” represent periods when keywords were more influential. Years in light green indicate that the keyword has not yet appeared, years in dark green indicate that the keyword has less influence, and years in red indicate that the keyword has more influence. The keywords Cluster Analysis (C). All of the keywords could be classified into seven categories, which were adenosine triphosphate, pulmonary fibrosis, nerve regeneration, tumor microenvironment, epitranscriptome, hypothermia, atherosclerosis. The timeline viewer of keywords.The timeline viewer of keywords (D).

Supplemental Information 6 Publications list of WOS.

Supplemental Information 7 Publications list of PubMed.

Supplemental Information 8 Publications list of Scopus.

We would like to thank all participants who volunteered their time to participate in the study.

Additional Information and Declarations

Competing Interests

Author Contributions

Data Availability

The authors declare that they have no competing interests.

Zewen Li conceived and designed the experiments, performed the experiments, analyzed the data, prepared figures and/or tables, authored or reviewed drafts of the article, and approved the final draft.

Yongfeng Lao conceived and designed the experiments, performed the experiments, prepared figures and/or tables, authored or reviewed drafts of the article, and approved the final draft.

Rui Yan conceived and designed the experiments, performed the experiments, prepared figures and/or tables, authored or reviewed drafts of the article, and approved the final draft.

Xin Guan analyzed the data, authored or reviewed drafts of the article, and approved the final draft.

Yanan Bai analyzed the data, authored or reviewed drafts of the article, and approved the final draft.

Fuhan Li analyzed the data, authored or reviewed drafts of the article, and approved the final draft.

Zhilong Dong conceived and designed the experiments, authored or reviewed drafts of the article, and approved the final draft.

The following information was supplied regarding data availability:

This is a literature review.

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
