# Peer review of "N6-methyladenosine (m6A) modification in inflammation: a bibliometric analysis and literature review"

_PeerJ, doi:10.7717/peerj.18645_

## Round 0.1 · original submission · Major Revisions

Please address issues pointed by the reviewers and amend manuscript accordingly

·

Basic reporting

The English language should be improved to ensure that an international audience can clearly understand your text.

Experimental design

The articles are reviewed properly, but it would be nice to include some recent findings which talk about m6A modification during viral infections and its impact on inflammation.

Validity of the findings

Impact and novelty were not assessed properly.

Additional comments

The study compiles findings on the role of m6A modifications in inflammation, specifically examining how m6A modifications influence processes such as apoptosis, autophagy, oxidative stress, and immune cell dysfunction.

Comments:
1. The introduction requires additional details on m6A writers, erasers, and readers.
2. The introduction section should emphasize the prevalence of m6A modifications in different regions of RNA, such as the 3’UTR, 5’UTR, or in exon/intron regions.
3. The authors should include a discussion on how viral infections influence cellular RNA m6A modifications and their subsequent control of the inflammatory response.
4. Protein names should be spelt out throughout the manuscript.
5. In the section on m6A and apoptosis, discuss whether BCL2, Bcl-xl, or Caspase mRNAs are modified by m6A in METTL3-depleted cells. Additionally, specify the cell type used in the referenced studies to clarify cell-type-specific responses.
6. Include details on the mechanism of the m6A/YTHDF1/BCL2 signaling axis (Line 214).
7. Discuss the contradictory roles of METTL3 in enhancing or reducing apoptosis.
8. Spell out YAP1 and describe its function in the context of apoptosis (Line 231).
9. Spell out ULK1 (Line 267).
10. At the end of the m6A and autophagy section, explain the possible reasons for the contradictory roles of m6A in autophagy.
11. Define NAC and its function (Line 290).
12. Spell out SOD, CAT, GSH-PX, and DGCR8.
13. Briefly define the functional role of miR-515-5p/TLR4/MyD88 in the inflammatory response (Line 475), and GPC4 (Line 477) for better reader understanding.
14. Change "m6A formation" to "m6A modification" (Line 512).
Suggestions:
It would be beneficial to emphasize the mechanisms by which m6A modifications on specific RNAs influence the expression of pro-apoptotic and autophagy-related proteins.

Reviewer 2 ·

Basic reporting

• The article combines a bibliometric analysis with a literature review, which is valuable for readers looking to understand both trends and the current state of knowledge on m6A and inflammation. The literature review is well-structured and covers critical mechanisms of m6A in inflammation, including apoptosis, autophagy, oxidative stress, and immune dysregulation.

• The list of publications analyzed in the bibliometric study should be included as a supplementary file to increase transparency and reproducibility.

• Figures need to be clearer and more reader-friendly. The authors should improve figure resolution and ensure that text within the figures is legible, particularly in Figure 6B.

• If Biorender or another application was used to generate Figures 8-11, proper citation should be provided as per journal guidelines.

Experimental design

• I thank the authors for acknowledging the limitations of their study (Lines 556-568). However, claiming that “our study may not fully reflect the actual trends in this field” (Lines 563-564) conflicts with one of the study’s primary objectives. To address this concern, I recommend verifying the robustness of the conclusions by cross-checking the results using additional databases, such as Scopus or PubMed. This will help to confirm if the trends identified from WoSCC are consistent across different databases.

• The inclusion of social sciences indexes in the bibliometric analysis is questionable, given the biological focus of the study. The authors should either justify this decision or revise their approach to exclude irrelevant sources and focus on biomedical databases.

• It is suggested that the authors compare their findings with bibliometric studies from related fields, such as cancer (PMC10906179) and viral infections (PMC10797797). This would provide valuable context for the relevance of the findings. Additionally, conducting keyword analyses across these related fields would offer insights into broader trends and gaps.

Validity of the findings

• The statement that no bibliometric studies of m6A in the context of inflammation exist (Lines 64-65) correctly highlights a gap in the field. However, it is essential to provide further depth to the analysis by discussing what conclusions can be drawn from the identified trends. For example, what are the current gaps in the field, and which areas require more research attention? By doing so, the study can offer more actionable insights for future investigations.

• The authors noted the exponential growth in m6A-related publications since 2018 (Lines 110-116). The authors could expound on the possible reasons behind this growth, such as advancements in m6A sequencing technologies or pivotal discoveries that created new paradigms in the field. Highlighting key landmark studies that catalyzed this growth would provide valuable context.

• Emerging trends, such as the potential for m6A-related therapies, should be discussed in more detail. This would not only enhance the review but also emphasize the practical implications of the research.

Additional comments

In conclusion, the manuscript provides a comprehensive bibliometric analysis and a thorough literature review of the current state of m6A research in inflammation. However, addressing the issues of database justification, incorporating more critical analysis of emerging trends, and verifying the findings with additional databases will significantly strengthen the manuscript. Once these suggestions are addressed, the study would provide more robust conclusions and be more aligned with the goals of bibliometric analysis, making it well-suited for publication in PeerJ.

---

## Round 0.2 · Minor Revisions

As you can see, both reviewers are satisfied by the scientific content of your revised manuscript. However, one of the reviewers indicated that the manuscript contains linguistic issues and requires careful proofreading by a native English speaker. Please note that PeerJ does not provide editing services, and therefore this task should be conducted by you.

·

Basic reporting

The paper properly reports the previous findings and connect with the recent trends.

Experimental design

N/A

Validity of the findings

The findings are addressed properly.

Additional comments

The article needs to be checked by native English speakers. I would recommend authors to read line by line and make necessary grammatical changes.

Reviewer 2 ·

Basic reporting

The authors have made the necessary suggested revisions in the manuscript, especially in enhancing the critical analysis within the discussion section to fully utilize insights from this bibliometric analysis. They also provided the necessary supplementary files with the list of publications used in their analyses, which improves the transparency and reproducibility of their study.

Experimental design

The authors have improved the details written in the methodology and clarified their choice of use of WOS database. They also provided a list of m6A-related references collated from other databases such as Scopus and PubMed. Of these, Scopus provided the most comprehensive list of publications. Could the authors confirm whether similar conclusions would be reached if Scopus or PubMed alone were used?

This reviewer also acknowledges the cross-comparison with previous bibliometric studies in related fields, such as m6A in cancer and viral infections. However, further critical analysis of these cross-comparisons would enhance the discussion, providing greater context for why these comparisons were performed and what they reveal about the positioning of m6A research within the broader scientific landscape.

Validity of the findings

This reviewer appreciates the improvements to the discussion section, particularly on how the authors outline the implications of their study in helping address knowledge gaps in m6A research. The extended discussion on the factors contributing to the recent growth in m6A-related publications and potential clinical applications are useful improvements in this revised manuscript. This expanded context provides readers with valuable insights into both fundamental and applied research trends in the field.

Additional comments

In conclusion, the authors have addressed the main points raised in the review, particularly by clarifying database choices, enhancing critical comparisons, and expanding the discussion on research trends and clinical implications of m6A. The above minor revisions are suggested to further refine the cross-comparative analysis with related bibliometric studies for a more impactful conclusion, but the paper is otherwise ready for publication.

---

## Round 0.3 · accepted · Accept

All remaining concerns were adequately addressed, and the revised manuscript is acceptable now.